# Highly selective nickel-catalyzed *gem*-difluoropropargylation of unactivated alkylzinc reagents

Lun An[1], Chang Xu[1] & Xingang Zhang [1]

In spite of the important applications of difluoroalkylated molecules in medicinal chemistry, to date, the reaction of difluoroalkylating reagents with unactivated, aliphatic substrates through a controllable manner remains challenging and has not been reported. Here we describe an efficient nickel-catalyzed cross-coupling of unactivated alkylzinc reagen\ts with *gem*-difluoropropargyl bromides. The reaction proceeds under mild reaction conditions with high efficiency and excellent regiochemical selectivity. Transformations of the resulting difluoroalkylated alkanes lead to a variety of biologically active molecules, providing a facile route for applications in drug discovery and development. Preliminary mechanistic studies reveal that an alkyl nickel intermediate [Ni(tpy)alkyl] (tpy, terpyridine) is involved in the catalytic cycle.

[1] Key Laboratory of Organofluorine Chemistry, Shanghai Institute of Organic Chemistry, University of Chinese Academy of Sciences, Chinese Academy of Sciences, 345 Lingling Road, Shanghai 200032, China. Correspondence and requests for materials should be addressed to X.Z. (email: xgzhang@mail.sioc.ac.cn)

The important role of organofluorinated compounds in medicinal chemistry, agrochemistry, and materials science[1, 2] has triggered substantial efforts in developing new and efficient methods for the controllable introduction of fluoroalkyl groups into organic molecules[3, 4]. Particularly, the preparation of difluoroalkylated compounds have received much attention over the past few years[5–7] due to the unique properties of difluoromethylene group (CF$_2$) that can improve the metabolic stability of biologically active molecules and lead to enhanced acidity of its neighboring group, increased the dipole moments as well as conformational changes[8, 9]. To date, the transition-metal-catalyzed difluoroalkylations have emerged as a useful strategy for the construction of Ar–CF$_2$R bond and thus to access difluoroalkylated arenes efficiently[6, 7, 10–14]. However, it remains challenging to adapt this strategy to the formation of Csp$^3$–CF$_2$R bond[15]. To the best of our knowledge, the reaction of difluoroalkylating reagents with unactivated, aliphatic substrates catalyzed by transition-metal has not been reported so far. Although some approaches to difluoroalkylated alkanes exist, they mainly rely on the transformation of functional groups, such as carbonyl, olefin, and alkyne[16–20]. We envisioned that if the transition-metal-catalyzed direct introduction of a CF$_2$ group into aliphatic chain at any position is successful, it would be a significant advance and provide an effective and general method for site-selective synthesis of difluoroalkylated alkanes.

Accordingly, herein, we demonstrate the feasibility of transition-metal-catalyzed cross-coupling of readily available alkylzinc reagents with gem-difluoropargyl bromides. The use of gem-difluoropropargyl bromides is because of their versatile synthetic utility of the carbon–carbon triple bond and their ready availability[21, 22]. Thus, this type of precursor would provide attractive and useful difluoroalkylated structures for applications in medicinal chemistry and materials science. However, efficient methods to such a valuable structure are very limited[15, 23]. As part of our ongoing study in transition-metal-catalyzed fluoroalkylations[11, 12, 24–26], herein, we describe a highly selective Ni-catalyzed cross-coupling between unactivated alkylzinc reagents and gem-difluoropropargyl bromides[27–31]. This method has several advantages: broad substrate scope, including a variety of unactivated alkylzinc reagents and gem-difluoropropargyl bromides; high efficiency and excellent regiochemical selectivity for the formation of gem-difluoropropargylated products; a catalyst loading of NiCl$_2$ or Ni(dppf)Cl$_2$ (dppf, 1,1′-bis(diphenylphosphino)ferrocene) as low as 0.5–5 mol%; and practical for gram-scale production.

## Results

**Optimization of the Ni-catalyzed cross-coupling.** We began our initial studies on the cross-coupling of 3-phenylpropylzinc bromide **1a** with (3-bromo-3,3-difluoroprop-1-yn-1-yl)triisopropylsilane **2a** in the presence of nickel catalyst with a variety of diamine ligands (Table 1, entries 1–8). However, only low yields of desired product **3a** were obtained. To our delight, switching diamine ligand with a triamine ligand (tpy) dramatically improved the yield of **3a** and no allenic side product was observed (entry 9). The beneficial effect of tpy on the current nickel-catalyzed process is in sharp contrast to previous work, in which diamine ligands were the best ligands for nickel-catalyzed difluoroalkylations[24, 32, 33]. We also found that the reaction was

---

**Table 1 Representative results for optimization of Ni-catalyzed cross-coupling of 1a with 2a[a]**

| Entry | [Ni] (x) | L (y) | 3a, Yield (%)[b] | Entry | [Ni] (x) | L (y) | 3a, Yield (%)[b] |
|---|---|---|---|---|---|---|---|
| 1 | NiCl$_2$·DME (10) | bpy (10) | 27 | 11 | NiBr$_2$·DME (10) | tpy (10) | 98 |
| 2 | NiCl$_2$·DME (10) | Phen (10) | 24 | 12 | Ni(PPh$_3$)Cl$_2$ (10) | tpy (10) | 91 |
| 3 | NiCl$_2$·DME (10) | **L1** (10) | 18 | 13 | Ni(dppb)Cl$_2$ (10) | tpy (10) | 95 |
| 4 | NiCl$_2$·DME (10) | **L2** (10) | 22 | 14 | Ni(dppf)Cl$_2$ (10) | tpy (10) | 99 |
| 5 | NiCl$_2$·DME (10) | **L3** (10) | 6 | 15 | Ni(dppf)Cl$_2$ (5) | tpy (10) | 98 |
| 6 | NiCl$_2$·DME (10) | **L4** (10) | 5 | 16 | Ni(dppf)Cl$_2$ (5) | tpy (5) | 98 |
| 7 | NiCl$_2$·DME (10) | **L5** (10) | 2 | 17[c] | Ni(dppf)Cl$_2$ (2.5) | tpy (2.5) | 99 (94)[d] |
| 8 | NiCl$_2$·DME (10) | **L6** (10) | 12 | 18[c] | Ni(dppf)Cl$_2$ (2.5) | tpy′ (2.5) | 90 |
| 9 | NiCl$_2$·DME (10) | tpy (10) | 82 | 19[c] | none | tpy (2.5) | nr |
| 10 | NiCl$_2$ (10) | tpy (10) | 89 | 20[c] | Ni(dppf)Cl$_2$ (2.5) | none | 9 |

TIPS triisopropylsilyl, DME dimethoxyethane, DMA dimethylacetamide
[a]Reaction conditions (unless otherwise specified): **1a** (0.45 mmol, 1.5 equiv in 1 mL of DMA), **2a** (0.3 mmol, 1.0 equiv), dioxane (2 mL)
[b]Determined by $^{19}$F NMR using fluorobenzene as an internal standard
[c]Using 1.2 equiv of **1a**
[d]Number in parenthesis is an isolated yield

**Table 2 Ni-catalyzed cross-coupling of unactivated alkylzincs 1 with 2a[a]**

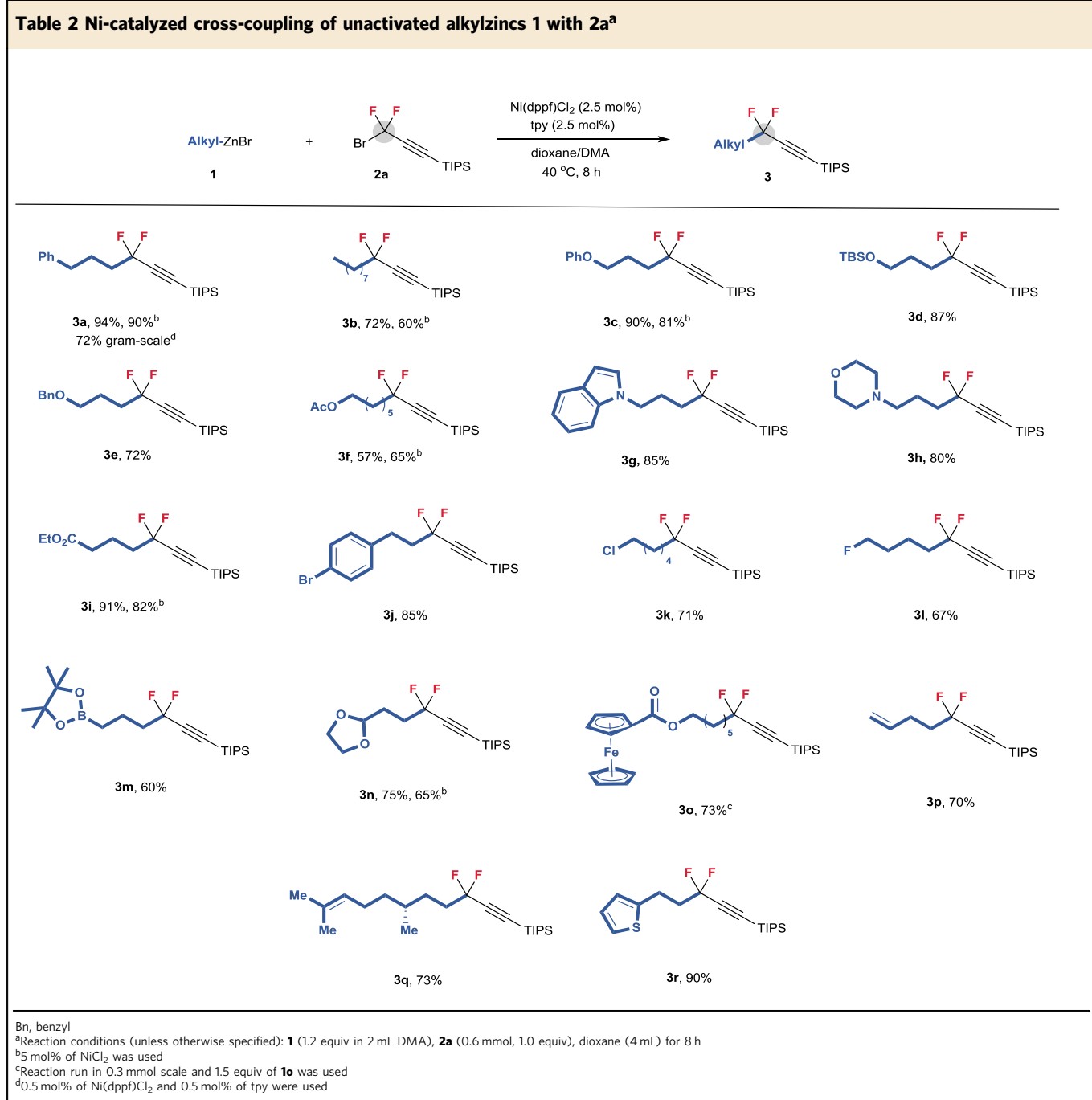

Bn, benzyl
[a]Reaction conditions (unless otherwise specified): **1** (1.2 equiv in 2 mL DMA), **2a** (0.6 mmol, 1.0 equiv), dioxane (4 mL) for 8 h
[b]5 mol% of NiCl$_2$ was used
[c]Reaction run in 0.3 mmol scale and 1.5 equiv of **1o** was used
[d]0.5 mol% of Ni(dppf)Cl$_2$ and 0.5 mol% of tpy were used

insensitive to the nature of nickel catalysts (entries 10–14). All the tested nickel salts showed good reactivities and Ni(dppf)Cl$_2$ (2.5 mol%) was the best choice, providing **3a** in 94% yield upon isolation (entry 17). Alternatively, the use of 4,4′,4″-tri-*tert*-butyl-terpyridine (tpy′) as a ligand also produced **3a** in a comparable yield (entry 18), thus providing a complementary ligand of choice for this process. No product was obtained in the absence of nickel catalyst (entry 19) and only 9% yield of **3a** was produced without ligand (entry 20), thus demonstrating the essential role of both nickel catalyst and ligand in promotion of the reaction.

**Scope of the Ni-catalyzed cross-coupling.** To ascertain the substrate scope of this method, a wide range of alkylzinc reagents were examined (Table 2). Overall, good-to-high yields of

*gem*-difluoropropargylated alkanes **3** were provided under optimized reaction conditions; even NiCl$_2$ was used as a catalyst, high product yields were still observed (**3a**–**3c**, **3f**, **3i**, **3n**). The reaction exhibits high-functional group tolerance. Substrates **1**, bearing important functional groups, such as ether, ester, indole, morpholine, ferrocenyl, and vinyl, all underwent the reactions smoothly (**3d**–**3i**, **3n**–**3q**). Importantly, the successful formation of **3j**–**3l** in good yields with intact bromide, chloride, and fluoride provides good synthetic handles for downstream transformations. Most remarkably, an alkylboronic acid ester was also compatible with the reaction conditions (**3m**), thus highlighting the high chemoselectivity of current nickel-catalyzed process. Moreover, thienyl-containing substrate furnished corresponding product **3r** with high efficiency (90% yield). The reaction was readily scalable as demonstrated by the gram-scale synthesis of **3a** (72% yield)

**Table 3 Ni-catalyzed cross-coupling of 1 with 2ᵃ**

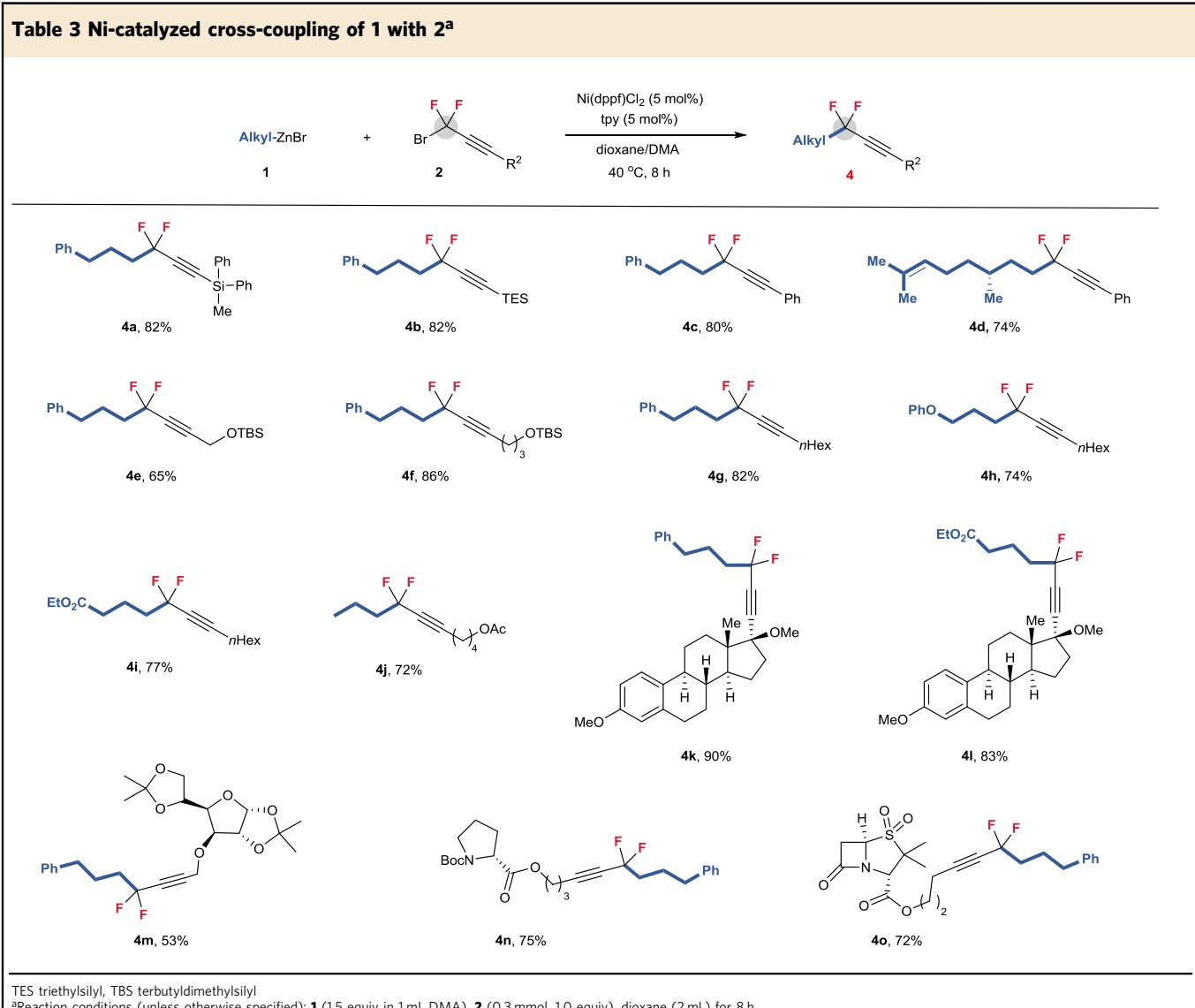

TES triethylsilyl, TBS terbutyldimethylsilyl
ᵃReaction conditions (unless otherwise specified): **1** (1.5 equiv in 1 mL DMA), **2** (0.3 mmol, 1.0 equiv), dioxane (2 mL) for 8 h

even when 0.5 mol% of Ni(dppf)Cl$_2$ was used, thus demonstrating the reliability of current process.

In addition to alkylzinc reagents, a variety of *gem*-difluoropropargylated bromides **2** were investigated (Table 3). Compounds **2** bearing silyl substituents, such as methyldiphenylsilyl and triethylsilyl groups, underwent the reaction smoothly (**4a** and **4b**). Aryl substituted **2** was also applicable to the reaction without observation of allenic side products (**4c** and **4d**). Notably, no defluorination side reaction occurred during the reaction process, in sharp contrast to previous reports, where defluorinations were the major pathway[34–37]. Remarkably, the alkyl substituted *gem*-difluoropropargylated bromides **2** were also competent coupling partners and provided corresponding products in good yields (**4e–4j**), in which the application of this method could efficiently lead to compound **4j**, a key intermediate for the preparation of bioactive molecule used for pheromone study[38]. To demonstrate the generality of this approach further, some biologically active molecules containing **2** were also examined. The estrone-derived substrate underwent the reactions smoothly (**4k** and **4l**), even 90% yield of **4k** was obtained. The glucose- and proline-containing *gem*-difluoropropargyl bromides were also amenable to the reaction (**4m** and **4n**). Because carbohydrates and amino

acids play important roles in life science, it would provide good opportunity to discover some interesting biologically active molecules. Importantly, sulbactam, a drug for the treatment of bacterial infection, derived substrate furnished **4o** efficiently, thus featuring the applicability of this protocol in drug discovery and development.

**Transformations of compounds 3 and 4.** The utility of this approach can also be highlighted by the transformations of resulting *gem*-difluoroalkylated alkanes. As shown in Fig. 1a, deprotection of **3a** with TBAF, followed by [3 + 2] cyclization[39] afforded difluoroalkylated pyrrole **7** in high yield. Sonogashira reaction of **5** with heteroaryl iodides proceeded smoothly, offering a complementary method to the approach illustrated in Table 3. The CF$_2$-containing alkane **10** can also be readily prepared through hydrogenation of carbon–carbon triple bond without defluorination, providing a facile route for site-selective introduction of a CF$_2$ group into an aliphatic chain (Fig. 1b). Notably, compounds **3** and **4** can serve as key intermediates for the synthesis of biologically active molecules. For example, compound **11**, a probe for the study of hydrophobic interaction in pheromone reception[38], was readily prepared by selective

reduction of compound **4f** (Fig. 1c). In light of a series of pheromone analogs containing Z-difluoroalkylated alkenes[38, 40], this transformation is highly relevant to the synthesis of such a kind of structures. Furthermore, compound **12** derived from *gem*-

difluoroalkylated alkane **3i** can also functionalize as a key intermediate for the synthesis of analog of omega-3 fatty acids **13** for therapeutic use (Fig. 1d)[41], in which the $CF_2$ group at propargylic position can improve the stability of omega-3 fatty acids

**Fig. 1** Transformations of compounds **3** and **4**. **a** Transformation of **3a**. Conditions: (i) TBAF, THF, −78 °C to −20 °C; (ii) $Ag_2CO_3$ (10 mol%), $EtO_2CCH_2NC$ **6**, dioxane, 80 °C; (iii) $Pd(PPh_3)Cl_2$ (5 mol%), CuI (5 mol%), HetArI **8**, $iPr_2NH$. **b** Transformation of **4i**. Conditions: (iv) Pd/C (10%), $H_2$ (1 atm). **c** Synthesis of pheromone derivative **11**. Conditions: (v) Lindlar catalyst, $H_2$ (1 atm). **d** Transformation of **3i**. Conditions: (vi) PBS (PH 7.4), 37 °C. **e** Synthesis of key intermediate **17** of biologically active molecule $PGF_{2a}$ derivative. Conditions: (vii) $Ni(dppf)Cl_2$ (5 mol%), tpy (5 mol%), *n*-Pentylzinc bromide, dioxane/DMA, 40 °C, 12 h

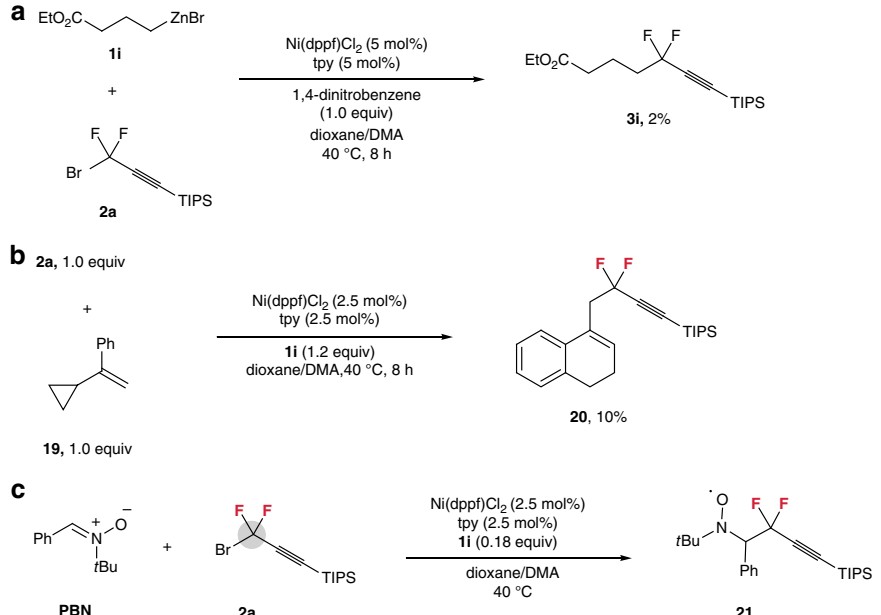

**Fig. 2** *Gem*-fluoropropargyl radical trapping experiments. **a** Radical-inhibition experiment. **b** Radical clock experiment. **c** EPR experiment

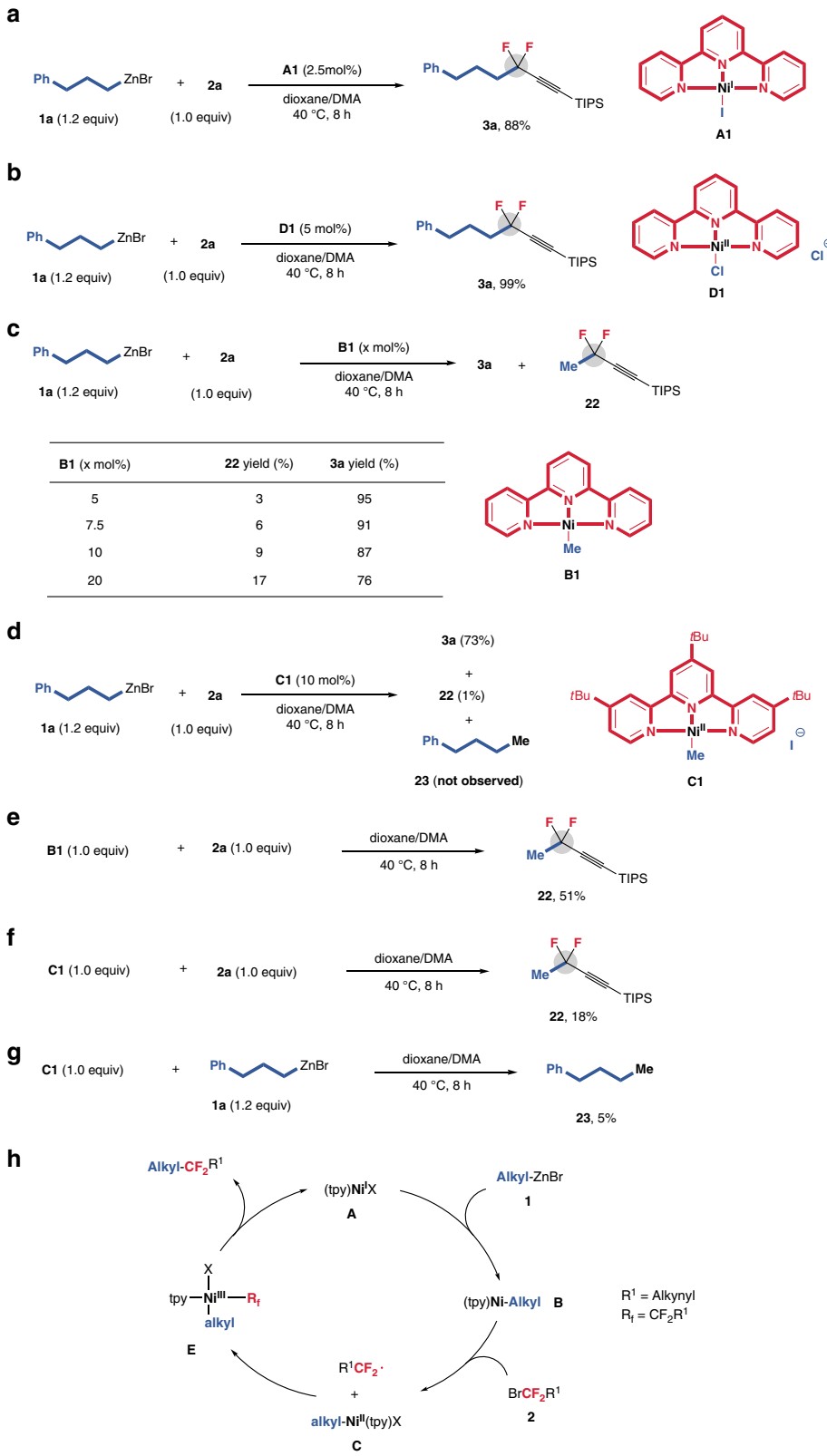

**Fig. 3** Mechanistic studies. **a** A1 catalyzed reaction of **1a** with **2a**. **b** D1 catalyzed reaction of **1a** with **2a**. **c** B1 catalyzed reaction of **1a** with **2a**. **d** C1 catalyzed reaction of **1a** with **2a**. **e** Stoichiometric reaction of **B1** with **2a**. **f** Stoichiometric reaction of **C1** with **2a**. **g** Stoichiometric reaction of **B1** with **1a**. **h** Proposed reaction mechanism

metabolites. Importantly, the copper-free click reaction[42] of terminal alkyne **12** with the dansyl fluorophore provided [3 + 2] cycloaddition products **15** with high efficiency (86%, as a mixture of regioisomers) in phosphate-buffered saline (PBS, pH 7.4) at 37 ºC (Fig. 1d). Yet, no reaction occurred by using its non-fluorinated counterpart (for details, see Supplementary Methods). Thus, these results clearly demonstrate that the presence of $CF_2$ group is critical for the high reactivity of **12** in the copper-free click reaction. This reactivity is not only because the lowest unoccupied molecular orbital (LUMO) of the difluoroalkylated alkyne can be lowered by the $CF_2$ group[43], but also because the stability of the transition state can be increased through hyperconjugative interactions between the carbon–carbon triple bond π system and vicinal $\sigma^{\star}_{C-F}$ acceptors[44]. Since compound **12** is stable and can be readily synthesized, such a kind of compound would have potential applications in chemical biology and materials science. Most remarkably, a key intermediate **17** for the synthesis of analog of prostaglandin $F_{2a}$ ($PGF_{2a}$) **18**, an eye drop for sterilization treatment[45, 46], was prepared efficiently through the current nickel-catalyzed process, thus demonstrating the utility of this approach further (Fig. 1e).

## Discussion

To gain mechanistic insights into the current reaction, radical-inhibition experiment was conducted. We found that the reaction was almost terminated (giving **3i** in a yield of 2%) by the electron transfer inhibitor (1,4-dinitrobenzene, 1.0 equiv)[47] that was added to the reaction mixture of **1i** and **2a** under standard reaction conditions (Fig. 2a). Furthermore, a ring-expanded product **20** (10% yield) was afforded when α-cyclopropylstyrene **19**, which is known to participate in radical rearrangements[48], was treated with **2a** under standard reaction conditions (Fig. 2b). This result confirms the possibility of forming *gem*-difluoropropargyl radical intermediate during the nickel-catalyzed process. As a direct evidence, the electron paramagnetic resonance (EPR) study of reaction of **2a** with spin-trapping agent phenyl *tert*-butyl nitrone[49] showed a spin adduct of the trapped *gem*-difluoropropargyl radical **21** ($a_N = 14.10$ G, $a_H^\beta = 2.01$ G) in the EPR sextet spectrum (Fig. 2c and Supplementary Fig. 168), suggesting that the difluoroalkyl group is substituted at the β-carbon atom of nitroxide so that the $a_N$ and $a_H^\beta$ values are decreased by the strong electron-withdrawing effect of $CF_2$ group through reducing the interaction of the electron spin with β-H atom as well as the electron pair on the nitrogen atom (usually, $a_N$ values are 14–16.5 G for the non-fluorinated dialkyl nitroxides)[49]. Moreover, EPR spectrum also showed the EPR signal of spin adduct **21** at the beginning of the reaction (Supplementary Fig. 169). Thus, a combination of the above results demonstrates that a *gem*-difluoropropargyl radical generated via a nickel-initiated single-electron transfer pathway is involved in the reaction.

We also prepared Ni(I) complex [(tpy)Ni$^I$-I][50, 51] (**A1**) and Ni(II) complex [(tpy)Ni$^{II}$Cl$_2$] (**D1**)[52] as the nickel species to add in the reaction of **1a** with **2a**, and found that both complexes catalyzed the reaction to afford compound **3a** with high yields (Fig. 3a, b), thus demonstrating that the ligand dppf in the nickel catalyst Ni(dppf)Cl$_2$ just play a trivial role to promote the reaction. We propose that dppf in this precatalyst benefits the formation of an active nickel species that can facilitate the catalytic cycle. To further investigate the detailed mechanism, we prepared methyl nickel complexes [(tpy)Ni-Me] (**B1**)[53] and [I-Ni(tpy′)Me] (**C1**)[54] as the catalysts in the mechanistic studies. In **C1** tpy′ is used instead of tpy because [I-Ni(tpy)Me] (**C1′**) is prone to decomposition to [(tpy)NiI][51], and because tpy′ as the ligand showed similar reactivity as tpy in the current cross-coupling

(Table 1, entry 18). In the presence of **B1** catalyst, the reaction could provide compound **3a** in high yield (Fig. 3c), as well as a methylated product **22** in a yield in the range of 3–17%, increasing as raising amount of **B1** from 5 to 20 mol% (Fig. 3c). On the contrary, the use of **C1** (10 mol%) as a catalyst led to only 1% yield of compound **22**, albeit formation of **3a** in a 73% yield (Fig. 3d). These findings suggest that compared with complex **C1**, complex **B1** is more favorable for the formation of **22**. In addition, cross-coupling product **23** was not generated between alkylzinc **1a** and **C1**, and even the stoichiometric reaction of **C1** with **1a** only afforded **23** in 5% yield (Fig. 3g). Both results indicate that a simple transmetalation of **C1** by **1a** is unlikely, ruling out a Ni(0/II) redox cycle. On the basis of the above analysis, we envisioned that a Ni(I/III) catalytic cycle is involved in the current reaction and alkyl nickel complex [(tpy)Ni-alkyl] (**B**) may be the key intermediate. To further support this possible pathway, a stoichiometric reaction of **B1** with **2a** (1.0 equiv) was conducted and provided **22** in 51% yield (Fig. 3e). However, only 18% yield of **22** was obtained when **C1** was used (Fig. 3f). Thus, the reaction via a key Ni(I) intermediate [(tpy)Ni-alkyl] (**B**) generated by a transmetalation-first pathway between alkyzinc and nickel(I) complex [(tpy)Ni-X] (**A**) is reasonable.

On the basis of above results[55, 56], a plausible reaction mechanism involving a Ni(I/III) catalytic cycle is proposed (Fig. 3h). The comproportionation of in situ-generated Ni(0) from the reaction of Ni(II) with alkylzinc reagent and the remaining Ni(II) species generates the Ni(I) complex [(tpy)Ni$^I$-X] (**A**)[53], which subsequently produces alkyl nickel complex [TpyNi-Alkyl] (**B**) through transmetalation with alkylzinc. Because such a terpyridine-based nickel(I) complex **B** is actually a nickel(II) complex [(tpy$^{-1}$)Ni$^{II}$-alkyl][53], a one-electron, ligand-based redox pathway would be occurred in the reaction of **B** with *gem*-difluoropropargyl bromide **2** to generate **C** and difluoroalkyl radical[53]. Recombination of **C** with difluoroalkyl radical produces nickel(III) complex [Alkyl(Tpy)Ni$^{III}$(R$_f$)X] (**E**, $R_f = CF_2CCR^2$), which subsequently undergoes reductive elimination to deliver *gem*-difluoropropargylated alkane and regenerate **A** simultaneously.

In conclusion, we have demonstrated a Ni-catalyzed difluoroalkylation of unactivated aliphatic substrates. The reaction proceeds under mild reaction conditions and allows a variety of alkylzinc reagents and *gem*-difluoropropargyl bromides with high efficiency and excellent chemoselectivity. Importantly, most of the resulting *gem*-difluoropropargylated alkanes are unknown and can serve as versatile building blocks for the organic synthesis and medicinal chemistry. Preliminary mechanistic studies reveal that the transmetalation between Ni(I) and alkylzinc to generate alkyl nickel complex [alkylNi(tpy)] initiates the reaction, and a Ni(I/III) catalytic cycle is involved in the reaction. This mechanism is different from that of recently reported nickel-catalyzed Negishi arylation of proparylic bromides, which has been suggested through a bimetallic pathway[56]. Because of the synthetic utility of carbon–carbon triple bond and the unique properties of $CF_2$, which can significantly improve the metabolic stability of biologically active molecules, this reaction will have applications in life and materials science as well as prompt the research area in fluoroalkylation of unactivated, aliphatic compounds from low-cost fluoroalkyl halides. Further studies to other derivative reactions are now in progress in our laboratory.

## Methods

**General procedure for nickel-catalyzed cross-coupling**. To a 25 mL of Schlenk tube was added Ni(dppf)Cl$_2$ (2.5 mol%) and tpy (2.5 mol%). The tube was evacuated and backfilled with Ar for three times, then *gem*-difluoropropargyl bromide **2a** (0.6 mmol, 1.0 equiv) and 1,4-dioxane (4 mL) were added. The resulting mixture was stirred at room temperature for 5 min, alkylzinc reagent **1** (0.72 mmol, 1.2

equiv in 2 mL of DMA) was then added dropwise over a period of 5 min. The tube was screw capped and put into a preheated oil bath (40 $^{\circ}$C). After stirring for 8 h, the reaction mixture was cooled to room temperature and diluted with EtOAc (50 mL). The resulting mixture was filtered with a pad of cellite. The filtrate was washed with water (15 mL), the organic layer was dried over $Na_2SO_4$, filtered, and concentrated. The residue was purified with silica gel chromatography to give the pure product.

**Data availability**. The authors declare that all the data supporting the findings of this study are available within the paper and its supplementary information files. For the experimental procedures, see Supplementary Methods. For NMR analysis of the compounds in this article, see Supplementary Figs. 1–166.

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

## Acknowledgements

Financial support for this work was provided by the National Basic Research Program of China (973 Program) (No. 2015CB931900), the National Natural Science Foundation of China (21425208, 21672238, 21421002, and 21332010), the Strategic Priority Research Program of the Chinese Academy of Sciences (No. XDB20000000), and SIOC.

## Author contributions

X.Z. and L.A. conceived and designed the experiments. L.A. performed the experiments. L.A. conducted the mechanism studies. C.X. prepared some starting materials. L.A. analyzed the data. X.Z. wrote the paper. All authors discussed the results and commented on the manuscript.

## Additional information

**Competing interests:** The authors declare no competing financial interests.

