## [Peer Review File · Nature Communications]

Reviewer #1 (Remarks to the Author):

Comments:

Difluoromethylene group (CF₂) could improve the metabolic stability of bioactive molecules, enhance acidity of its neighboring group, increase dipole moments and lead to conformational changes. The CF₂ functionality plays an important role in organic chemistry, medicinal chemistry, agrochemistry as well as materials science. Several transition-metal-catalyzed methods for construction of Ar-CF₂R bond have been revealed, demonstrating a powerful and efficient strategy for preparation of aromatic CF₂-compounds; however, the conventional formation of Csp³-CF₂R was challenging and mainly depended on the functional group transformation from carbonyl, olefin and alkyne. In this manuscript, Zhang and co-workers disclosed the first example for realizing direct Csp³-CF₂R bond construction by nickel catalyzed coupling between alkylzinc reagents and gem-difluoropargyl bromides. This reaction reported here showed diverse functional groups tolerance, high efficiency and excellent regiochemical selectivity, low catalyst loading and good feasibility for potential useful downstream transformations and gram scale production. In addition, detailed preliminary mechanistic studies were designed and conducted, and the results showed that a single electron transfer process and a Ni(I/III) catalytic cycle may be involved in this transformation. The experiments were described carefully, and all of the compounds were characterized comprehensively.

The manuscript is well organized, and the results are encouraging. The topic of this manuscript meets the reader interest of the journal, and it will be suitable for publication in the journal. However, there are several mistakes or confusions in the manuscript, that need to be improved or be addressed.

One of the major concern is about the post CF₂-coupling functional group manipulation. While authors demonstrated a variety of transformation with alkyne in cycloaddition, click chemistry etc, the reduction to aliphatic chain hasn't been demonstrated yet to fully realize the potential of this methodology for CF₂-aliphatic formation. I hope this could be strengthened in the revised manuscript.

Minor comments.

1. In Table 1, entry 17, there is a number in parenthesis. Please make a note of it. I think it should be an isolated yield.
2. In Table 3, the reaction scheme, the product number should be 4, not 3.

3. Line 131-133, the gram-scale preparation of 3a appear in this section maybe not proper. It is not the "transformation of 3a". Removal it to the table 2 and corresponding text section should be more reasonable.
4. Line 31-33, the authors said "... if the transition-metal-catalyzed direct introduction of a CF₂ group into aliphatic chain at any position ...". Most of reactions in this manuscript were depicted as alpha-CF₂ of the alkyne functionality. In order to better illustrate this goal, the authors could add a transformation for the reduction of alkyne to alkene/alkane.
5. Line 136, "...to the approach illustrated in Scheme 3". Please double check whether it is Scheme 3.
6. Line 185, "The comproportionation of in situ generated Ni(0)..." The authors should explain how to generate Ni(0) species.
7. In ESI page S6, not 3b in the table.

Reviewer #2 (Remarks to the Author):

Zhang and co-workers describe the Ni-catalyzed cross-coupling of alkylzinc reagents and gem-difluoropropargyl bromides. Overall, this is a nice methodology study, but the authors do not give sufficient context for the importance of the products of the methodology. Why are these products important? What is the significance of the gem-difluoropropargyl functional group? Perhaps the products are bioactive or this functional group is represented in bioactive compounds. Without this context, I do not see the justification for publication in Nature Communications. With respect to the mechanistic work, the authors need to provide more context for their interpretation of these studies.

The authors should cite Koichi Mikami's work on gem-difluoropropargylation: *Org. Lett.* 2016, 18, 3354-3357

Comments on mechanistic studies:

The mechanistic proposal is certainly feasible, but as written, I do not find the results to be as conclusive as presented. Perhaps this issue could be resolved with more discussion about these studies.

The authors state that 'We found that the reaction almost shut down when a radical scavenger (TEMPO, 0.5 equiv) or an electron transfer inhibitor (1,4-dinitrobenzene, 1.0 equiv) was added to the reaction mixture of 1i and 2a under standard reaction conditions,' but they then show that TEMPO reacts with the propargyl bromide and that the process is Ni-catalyzed. Perhaps the reason that the reaction is almost shut down is because the propargyl bromide is being consumed. Are other products formed with dinitrobenzene? It is good practice to use multiple radical scavengers/inhibitors precisely because they are not always 'innocent'.

The authors then say that 'these results demonstrate that a gem-difluoropropargyl radical generated via 1a nickel initiated single-electron-transfer (SET) pathway is involved in the reaction, which was further confirmed by the EPR experiments (for details, see the Supporting Information).' The authors do not discuss the experiments, and merely state that they support the proposed mechanism. Perhaps they do, perhaps they don't. The SI shows a reaction between 2a (the propargyl bromide) and PBN – which I had to look up. PBN is a spin-trapping reagent. This needs to be discussed (I don't think this will be obvious to most readers).

The PBN reaction doesn't necessarily operate by the same mechanism as the title reaction. It might, but there's not enough evidence to say unequivocally that the PBN reaction is proof of a SET reaction for the cross-coupling. Did the authors monitor the reaction by EPR?

The authors also state that 'These findings clearly demonstrate that the complex (B1) instead of complex (C1) initiates the formation of 14.' B1 is certainly better at forming 14, but C1 does for 14 in low yields. Moreover, B1 and C1 are not directly analogous. Perhaps that is the reason for the lower yields with C1.

It's also not clear to me why there is such a change by using tpy.

Reviewer 1.

R1: In Table 1, entry 17, there is a number in parenthesis. Please make a note of it. I think it should be an isolated yield.

The note of Table 1, entry 17 has been added. The number in parenthesis is an isolated yield.

R2: In Table 3, the reaction scheme, the product number should be **4**, not **3**.

The product number 3 in Table 3 (currently, Table 3 is Fig. 3) has been changed into 4.

R3: Line 131-133, the gram-scale preparation of **3a** appear in this section maybe not proper. It is not the “transformation of **3a**”. Removal it to the table 2 and corresponding text section should be more reasonable.

The gram-scale synthesis of 3a has been moved to the Table 2 (currently, Table 2 is Fig. 2).

R4: Line 31-33, the authors said “... if the transition-metal-catalyzed direct introduction of a CF₂ group into aliphatic chain at any position ...”. Most of reactions in this manuscript were depicted as alpha-CF₂ of the alkyne functionality. In order to better illustrate this goal, the authors could add a transformation for the reduction of alkyne to alkene/alkane.

The transformation of gem-difluoropropargylated alkanes into alkene/alkane has been conducted, please see Fig. 4b-c.

R5: Line 136, “...to the approach illustrated in Scheme 3”. Please double check whether it is Scheme 3.

Scheme 3 should be Fig. 3 in current revised version.

R6: Line 185, “The comproportionation of in situ generated Ni(0)...” The authors should explain how to generate Ni(0) species.

The Ni(0) species was generated from the reaction of Ni(II) with alkylzinc reagent. We have added

this comment to the text in mechanistic studies section.

R7: In ESI page S6, not **3b** in the table.

The **3b** in the table in ESI page S6 should be **3i**. We have corrected it.

Reviewer 2

Zhang and co-workers describe the Ni-catalyzed cross-coupling of alkylzinc reagents and gem-difluoropropargyl bromides. Overall, this is a nice methodology study, but the authors do not give sufficient context for the importance of the products of the methodology. Why are these products important? What is the significance of the gem-difluoropropargyl functional group? Perhaps the products are bioactive or this functional group is represented in bioactive compounds. Without this context, I do not see the justification for publication in Nature Communications.

*We very much appreciate the recognition of the practicality of our methodology. To demonstrate the importance of the products **3** and **4** of the methodology, we have conducted additional experiments from **3** and **4** to prepare biologically active molecule **11** as well as the key intermediates **12** and **17** used for the synthesis of biologically active molecules (Fig. 4).*

*As shown in Fig. 4, firstly, the synthetic utility of carbon-carbon triple bond can make compounds **3** and **4** as versatile building blocks for the organic synthesis. For examples, the alkyne **3a** can be efficiently converted into heterocycle-containing compounds **7** and **9** through [3+2] cyclization or Sonogashira coupling (Fig. 4a). Reduction of **3a** can also lead to difluoroalkylated alkane **10** efficiently, thus providing a facile route for site-selective introduction of a CF₂ group into aliphatic chain (Fig. 4b).*

*Secondly, incorporation of CF₂ at propargylic or allylic position can significantly improve the metabolic stability of biologically active molecules, which has become a useful strategy for the modification of biologically active molecules. As shown in Fig. 4c, semi-reduction of **4j** efficiently provided difluoro pheromone derivative **11**, a probe for the study of hydrophobic interaction in pheromone reception (ref. 38). Actually, a series of difluoro derivative of the sex pheromone can be prepared through this method (see ref. 38 and 40). Furthermore, alkyne **12** derived from compound **3i** can serve as a key intermediate for the preparation of analogue of omega-3 fatty acids **13** for therapeutic use (Fig. 4d), in which the CF₂ group at propargylic position can improve the stability of omega-3 fatty acids metabolites. For the synthesis of **13** from compound **12**, please see ref. 41, in which compound **13** can be readily prepared through Sonogashira cross-coupling of **12** with E-vinyl iodide, followed by deprotection. Additionally, application of this method can efficiently prepare another intermediate **17** used for the synthesis of analogue of prostaglandin F_{2a} (PGF_{2a}) **18**, an eye drop for sterilization treatment. According to the ref. 43 and 44, compound **18** can be*

easily prepared from alkyne **17**.

Thirdly, the presence of CF₂ moiety at propargylic position not only can lower the LUMO of the difluoroalkylated alkyne (ref. 43), but also can increase the stability of the transition state (TS) of [3+2] cyclization reaction through hyperconjugative interactions between the carbon-carbon triple bond system and vicinal σ^*_{C-F} acceptors (ref. 44). Therefore, the products of the methodology would have potential applications in chemical biology through copper-free click reaction (see ref. 42). For example, compound **12** underwent the copper-free click reaction smoothly at 37 °C in PBS (PH 7.4) buffer (Fig. 4d).

Therefore, the products of the methodology are important for the organic synthesis and medicinal chemistry and our method would be useful in drug discovery and development.

The authors should cite Koichi Mikami's work on gem-difluoropropargylation: Org. Lett. 2016, 18, 3354-3357.

The Koichi Mikami's work on gem-difluoropropargylation: Org. Lett. 2016, 18, 3354-3357 has been cited, please see ref 15. This is a very nice contribution for the synthesis of gem-difluoropropargylated compounds.

Comments on mechanistic studies:

The mechanistic proposal is certainly feasible, but as written, I do not find the results to be as conclusive as presented. Perhaps this issue could be resolved with more discussion about these studies.

The authors state that 'We found that the reaction almost shut down when a radical scavenger (TEMPO, 0.5 equiv) or an electron transfer inhibitor (1,4-dinitrobenzene, 1.0 equiv) was added to the reaction mixture of **1i** and **2a** under standard reaction conditions,' but they then show that TEMPO reacts with the propargyl bromide and that the process is Ni-catalyzed. Perhaps the reason that the reaction is almost shut down is because the propargyl bromide is being consumed. Are other products formed with dinitrobenzene? It is good practice to use multiple radical scavengers/inhibitors precisely because they are not always 'innocent'.

The radical inhibition experiment with a radical scavenger (TEMPO) was removed from the text. However, the use of an electron transfer inhibitor (1,4-dinitrobenzene, 1.0 equiv) almost shut down the reaction with only 2% yield of cross-coupling product **3i** obtained (Fig. 5a). We did not observe any other byproducts formed with dinitrobenzene. For the reference of using dinitrobenzene as an electron transfer inhibitor, see ref. 47. In addition, we also did radical clock experiment (Fig. 5b).

A ring-expanded product **20** (10% yield) was afforded when α -cyclopropylstyrene **19**, which is known to participate in radical rearrangements (ref. 48), was treated with **2a** under standard reaction conditions. These findings demonstrate that the generation of a gem-difluoropropargyl radical species in the reaction is possible.

The authors then say that ‘these results demonstrate that a gem-difluoropropargyl radical generated via a nickel initiated single-electron-transfer (SET) pathway is involved in the reaction, which was further confirmed by the EPR experiments (for details, see the Supporting Information).’ The authors do not discuss the experiments, and merely state that they support the proposed mechanism. Perhaps they do, perhaps they don’t. The SI shows a reaction between **2a** (the propargyl bromide) and PBN – which I had to look up. PBN is a spin-trapping reagent. This needs to be discussed (I don’t think this will be obvious to most readers).

*The EPR experiment has been discussed in the text, see Fig. 5c. An EPR study of reaction of **2a** with spin-trapping agent phenyl tert-butyl nitron (PBN), in which a spin adduct of the trapped gem-difluoropropargyl radical **21** was produced (Fig. 5c). The EPR showed an EPR sextet spectrum of nitroxide **21** ($a_N = 14.10$ G, $a_H^\beta = 2.01$ G), suggesting that the difluoroalkyl group is substituted at the β -carbon atom of nitroxide and the a_N and a_H^β values are decreased by the strong electron-withdrawing effect of CF_2 group through reducing the interaction of the electron spin with β -H atom as well as the electron pair on the nitrogen atom (usually, for non-fluorinated dialkyl nitroxides, a_N values are 14-16.5 G, ref. 49).*

The PBN reaction doesn’t necessarily operate by the same mechanism as the title reaction. It might, but there’s not enough evidence to say unequivocally that the PBN reaction is proof of a SET reaction for the cross-coupling. Did the authors monitor the reaction by EPR?

*The use of EPR with PBN as a spin-trapping reagent for the study of SET reaction has been reported before (see ref. 49), in which a Pd(0) induced reaction of addition of fluoroalkyl iodides to alkenes was proved to be occurred via a SET pathway. Therefore, the use of PBN as a spin-trapping reagent to study the SET pathway in current reaction is feasible. We also monitored the reaction by EPR (0-25 min) and observed the EPR signal of spin adduct **21** at the beginning of the reaction (Supplementary Fig. 2). Thus, a gem-difluoropropargyl radical generated via a nickel initiated single-electron-transfer (SET) pathway in the reaction is reasonable.*

The authors also state that ‘These findings clearly demonstrate that the complex (**B1**) instead of complex (**C1**) initiates the formation of **14**.’ **B1** is certainly better at forming **14**, but **C1** does for **14** in low yields. Moreover, **B1** and **C1** are not directly analogous. Perhaps that is the reason for the lower yields with **C1**.

It’s also not clear to me why there is such a change by using tpy.

*The replacement of tpy with tpy’ in **C1** is because of the instability of [I-Ni(tpy)Me] (**C1**’), which is prone to decomposition and leads to [(tpy)NiI] (ref. 51). In addition, **C1** is stable at room temperature and tpy’ showed similar reactivity with tpy. When tpy’ was used as a ligand under standard reaction conditions, a comparable yield of **3a** was provided (Table 1, entry 18). Therefore, the use of **C1** instead of **C1**’ would not influence the reaction efficiency. We have added these comments into the text. The low yield of **14** (currently, **14** is **22**) from **C1** demonstrates that the complex (**B1**) instead of complex (**C1**) may initiate the formation of **22**. To further support the possible pathway that a Ni(I/III) catalytic cycle is involved in the current reaction and alkyl nickel complex [(tpy)Ni-alkyl] (**B**) may be the key intermediate, a stoichiometric reaction of **B1** with **2a** (1.0 equiv) was conducted and provided **22** in 51% yield (Fig. 6e). However, only 18% yield of **22** was obtained when **C1** was used (Fig. 6f). Thus, the reaction via a key intermediate [(tpy)Ni-alkyl] (**B**) generated by a transmetalation-first pathway between alkylzinc and nickel(I) complex [(tpy)Ni-X] (**A**) is reasonable.*

Reviewer #1 (Remarks to the Author):

All mistakes / discrepancy have been addressed in the revised manuscript. The organization is much clear and the mechanism study is sound. The application and usefulness of gem-difluoropropargylation have also been emphasized. I agree with the publication in Nature Communications.

Only one minor suggestion is to change the color of compound 4 in transformation scheme of figure 3 from red to black.

Reviewer #2 (Remarks to the Author):

I am satisfied that the authors made adequate changes to the manuscript based on all reviewer comments. I am pleased to recommend publication in Nature Communications.